# Effects of Cysteine on Physicochemical Properties of High-Moisture Extrudates Prepared from Plant Protein

**DOI:** 10.3390/foods11193109

**Published:** 2022-10-06

**Authors:** Huihui Dai, Hongzhou An

**Affiliations:** College of Food Science and Technology, Henan University of Technology, Zhengzhou 450001, China

**Keywords:** cysteine, plant protein, high-moisture extrusion, fiber structure, odor properties

## Abstract

The effects of cysteine addition (0%, 0.05%, 0.10%, 0.15%, 0.20%, 0.25%) on the physicochemical properties of plant-based extrudates by high-moisture extrusion were investigated. The texturization degree, rheological properties, hardness, springiness and chewiness of the extrudates significantly improved with the addition of cysteine (<0.15%). Analysis of the microstructure showed that the addition of cysteine (<0.15%) improved the formation of the fiber structure in the extrudates. Cysteine (<0.15%) promoted the formation of new disulfide bonds and non-covalent bonds by changing the disulfide bonds of protein molecules and enhanced the cross-linking degree between proteins. However, the excessive addition of cysteine (>0.15%) had a negative impact on the formation of fiber structure. Analysis of secondary structure suggested that the ordered β-sheet structures gradually converted to the disordered β-turn and random coil structures with the addition of cysteine (>0.15%). In addition, analysis of odor properties with the addition of cysteine using an electronic nose showed the difference in volatile components of the extrudates mainly existed in halides, hydrocarbons, sulfides and amines. On the whole, the addition of cysteine could improve the quality of plant-based extrudates.

## 1. Introduction

In recent decades, the production of animal meat has faced many pressures, including consuming a lot of natural resources, destroying the ecological environment, violating animal welfare and bringing health problems [1,2]. Therefore, people have started to explore meat substitutes as a solution [3,4]. However, as a healthy and nutritious food, plant-based meat substitutes can meet the needs of consumers and the sustainability of food supply in the future. In recent years, the market for plant-based meat substitutes has developed rapidly [5]. The development of plant-based meat substitutes requires the creation of texture, appearance, flavor and taste, similar to traditional meat products. At present, the main technologies for preparing the fiber structure of meat analogues are extrusion technology, freeze structuring, shearing technology, electrospinning and crosslinking [6]. In particular, extrusion technology is a better choice for the production of plant-based meat substitutes with fiber structure [7].

With the in-depth study of extrusion technology, high-moisture extrusion technology is widely used in the production of plant-based meat substitutes because of its rich fiber structure, high production efficiency, high nutrient retention and no waste discharge [8]. High-moisture extrusion technology is carried out under high moisture content (40~80%) and a long cooling die needs to be installed at the outlet of the extruder [9]. During the extrusion process, the plant protein and water are mixed and the material is subjected to the action of high temperature, high pressure and high shear in the barrel and the protein molecular chain expands and agglomerates [10]. Protein molecules rearranged and crosslinked in the cooling die, which contributed to the formation of fiber structure [11].

Currently, soybean protein and wheat gluten are the most important proteins in the production of plant-based meat substitutes [12,13]. Soybean protein has rich sources and excellent processing properties. It is rich in nutritional value and contains a variety of essential amino acids, which is comparable to animal-derived foods [14]. In addition, long-term intake of soy protein can prevent cardiovascular disease, improve the lipid metabolism and reduce cholesterol [15]. Wheat gluten easily absorbs water and stretches the structure to form a network structure, which has good viscoelasticity, extensibility and thermal coagulation. Elastic gel can be formed during the extrusion process, which facilitates the formation of fiber structure of the extrudates [3]. Studies have shown that the combination of two or more proteins is conducive to the formation of fiber structure [16]. Soy protein isolate is usually combined with wheat gluten to enhance the texture properties of the meat substitutes [17,18].

Previous studies have shown that the interaction between proteins is closely related to the fiber structure and the texture properties of the extrudates. Non-covalent bonds (hydrophobic interactions and hydrogen bonds) and disulfide bonds are important forces to maintain protein molecular cross-linking, contributing significantly to the formation of fiber structure [19,20]. At present, our research team is exploring a better compounding ratio for the production of high-moisture textured plant protein raw materials (wheat gluten and soy protein), but the extrudates still have poor fiber structure and texture characteristics. Therefore, additives need to be added to improve the quality of the extrudates. Cysteine is an amino acid containing reductive sulfhydryl groups. Under certain conditions, cysteine is oxidized to cystine to form disulfide bonds, which are relatively stable covalent bonds and play a role in stabilizing the spatial structure of peptide chains in protein molecules. Therefore, cysteine can affect the aggregation and cross-linking of protein molecules through disulfide bonding and then affect the texture properties of the extrudates [21,22]. In addition, because cysteine is a sulfur-containing amino acid, it will produce a smell similar to meat after heat treatment, which is very important for the formation of food aroma components [23,24]. So far, there are few studies on the structural and odor properties of cysteine on high-moisture extruded plant-based meat analogues. Therefore, an appropriate additional amount of cysteine can improve the quality of the extrudates by high-moisture extrusion.

This paper mainly studied the effect of cysteine addition (0%, 0.05%, 0.10%, 0.15%, 0.20%, 0.25%) on the physicochemical properties of the plant-based extrudates. The color, texturization degree, texture properties, microstructure, rheological properties, protein solubility, protein secondary structure and odor properties of the extrudates were determined to analyze the mechanism of cysteine on the quality of the extrudates.

## 2. Materials and Methods

### 2.1. Materials

Soy protein isolate and wheat gluten protein were provided by Yihai Kerry (Shanghai, China). Soy protein concentrate was purchased from Yuwang (Dezhou, Shandong, China). Cysteine was provided by Huayang Biotechnology Co., Ltd. (Hengshui, Hebei, China). In this experiment, the materials were food grade and the chemical reagents were analytical grade.

### 2.2. Extrusion Experiments

All extrusion experiments were carried out using a twin-screw extruder (CLEXTRAL Ev025, Clestero Co., Ltd., Firminy, France), equipped with two screws with a screw diameter of 25 mm and a length of 600 mm. The extruder was divided into feeding zone, mixing zone, cooking zone and cooling zone. A long cooling die attached to the end of the extruder was equipped with cooling circulating water and the cooling temperature was 60 °C. The cross-sectional size of the cooling die was 30 mm × 4 mm. The mass ratio of soy protein and wheat gluten protein was 7:3. The protein powders and cysteine were configured according to mass ratios (0%, 0.05%, 0.10%, 0.15%, 0.20% and 0.25% *w*/*w*) before feeding. The operating parameters of the extruder were screw speed 280 rpm, feeding speed 3.4 kg/h and material moisture content 53%. The extruder barrel had six temperature control sections from I to VI were 30 °C, 90 °C, 120 °C, 140 °C, 150 °C and 160 °C. The extrudates were collected after the extruder worked stably, vacuum sealed and frozen at −18 °C until use.

### 2.3. Color Determination

The color of the extrudates was determined by a colorimeter (CR-400, Konica Minolta, Tokyo, Japan). The value of lightness (*L**), redness (*a**), yellowness (*b**) and Δ*E** of the extrudates were obtained, where Δ*E** represents the color difference between the extrudates and the white board [25].

### 2.4. Texturization Degree Determination

The extrudates were cut into a shape of 2 cm × 2 cm, where the transverse and longitudinal shear force were determined by a physical property tester (TA-XT Plus, Stable Micro Systems, Godalming, Surrey, UK) and the ratio represented the texturization degree of the extrudates. According to previous research methods [26,27], the operating parameters were shearing degree 75%, speed before measurement 2.0 mm/s, speed during measurement 1.0 mm/s and speed after measurement was 2.0 mm/s.

### 2.5. Texture Properties Determination

The texture analyzer (TA-XT Plus, Stable Micro Systems, UK) was used to determine the texture properties of the extrudates, including hardness, elasticity and chewiness. The determination was performed following the previous research method [28]. A piece of the extrudates with 2 cm × 2 cm was compressed using a P/36R probe to 75% under the texture profile analysis (TPA). The parameters were speed before determination 2.0 mm/s, speed during determination 1.0 mm/s and speed after determination 2.0 mm/s.

### 2.6. Microstructure Observation

The microstructure observation of the lyophilized extrudates was conducted using a scanning electron microscope (Quanta250FEG, FEI Company, Portland, OR, USA). The preprocessing method was based on the previous research with some modifications [29]. The extrudates were cut into small pieces with 7 mm × 7 mm × 4 mm. The extrudates were soaked in glutaraldehyde (2.5%, *v*/*v*) for 24 h and then washed with 0.1 M phosphate buffer 3 times. The extrudates were eluted with 30%, 60%, 90% and 100% ethanol for 15 min. The cross section of the extrudate was placed in a scanning electron microscope and magnified 3000-times for observation.

### 2.7. Rheological Properties Determination

The rheological behaviors of the extruded samples within the linear viscoelastic region were examined using a rheometer (HAAKE RS6000, Thermo Fisher Haake Corporation, Waltham, MA, USA). Thus, 4 g powder of the extruded samples was dispersed into 13 mL distilled water, mixed with a vortex shaker for 60 s, fully hydrated for 1 h at room temperature and stored at 4 °C overnight. A strain sweep test was performed to determine the linear viscoelastic region, for which the strain was determined to be fixed at 0.2% according to the constant G* was a linear viscoelastic region. The frequency sweep was measured in a range of 0.1–100 Hz at 25 °C. The storage modulus G′ and loss modulus G″ of the dispersed samples were recorded.

### 2.8. Protein Solubility Determination

The protein solubility of the extrudates was analyzed to assess the protein–protein interactions during high-moisture extrusion. The determination method was based on the previous research [30]. The extraction solutions were chosen as follows: (1) 0.035 mol/L phosphate buffer with pH 7.6 (PB); (2) 8 mol/L urea in the PB (PB + Urea); (3) 1.5 g/100 mL sodium dodecyl sulfate (SDS) in the PB (PB + SDS); (4) 0.1 mol/L 2-mercaptoethanol (2-ME) in the PB (PB + 2-ME). The extrudate powders were dissolved in the above two solutions, respectively. The total protein content of the extrudates was measured using the Kjeldahl method (Kjeltec 8400, Foss, Copenhagen, Denmark). The soluble protein content in the supernatant was determined by the Coomassie Brilliant Blue method at 595 nm (UV762, Shuangxu Electronics Co., Ltd., Shanghai, China) using bovine serum albumin (BSA) powder as a standard. Protein solubility was calculated as the ratio of the supernatant and total protein in the extrudates. The content of disulfide bonds was determined based on the differences in protein content between solutions PB and PB + 2-ME, respectively.

### 2.9. Fourier Transform Infrared Spectroscope Determination

The secondary structure content of the extrudates was analyzed by Fourier transform infrared spectroscopy (NICOLET 6700, Thermo Fisher Scientific, Waltham, MA, USA). The determination method was based on the previous research [31]. The extrudate powders were mixed with KBr at a ratio of 1:100. The operating parameters were spectral wavenumber range 400–4000 cm^−1^, resolution 4 cm^−1^ and scan 64 times. Omnic (version 8.0, Thermo Nicolet, Waltham, MA, USA) was used to analyze each spectrum (1600–1700 cm^−1^) and Peakfit (version 4.12, SPSS, Chicago, IL, USA) was used for gaussian fitting and second-order derivation to analyze secondary structure changes.

### 2.10. Electronic Nose Measurement

The aroma analysis of extrudates was performed using a Fox 4000 electronic nose (Alpha M.O.S., Toulouse, France) equipped with 17 metal-oxide semiconductors. The determination method was based on the previous research with minor modifications [32,33]. The extrudates were pulverized in liquid nitrogen and 2.5 g of the sample was accurately weighed into a 10 mL special headspace bottle for test. The measurement parameters were headspace heating temperature 70 °C, heating time 200 s and sample injection volume 500 μL.

### 2.11. Statistical Analysis

The data were analyzed using SPSS statistical software (IBM, New York, NY, USA). Significance analysis was determined by one-way analysis of variance (ANOVA), followed by Duncan’s method to evaluate significant differences (*p* < 0.05).

## 3. Results and Discussion

### 3.1. Color of the Extrudates

As shown in Table 1, with increasing cysteine content from 0% to 0.25%, the color of the extrudates indicated a decrease in the *a** (from 7.11 to 6.89) value but an increase in the *b** (from 11.45 to 12.66) and Δ*E** (from 39.92 to 41.46) values. However, there was no significant difference in the change in the *L** value (from 60.77 to 59.61).

Moreover, Δ*E** represents the color difference between the tested product and the standard whiteboard. The Δ*E** value increased significantly with the addition of cysteine, indicating that the color of the extrudates was significantly deepened and darkened. It might because the addition of cysteine could promote the Maillard reaction in the barrel of the extruder. Previous studies believed that the color changes in plant-based extrudates were mainly caused by Maillard reaction during extrusion [34,35]. We could infer that the higher addition of cysteine (>0.15%) resulted in darker color of the extrudates.

### 3.2. Texturization Degree and Texture Properties

The texturization degree referred to the ratio of the shear force in the vertical direction and the shear force in the parallel direction was an important index to indicate the formation of fiber structure of the extrudates [26]. As shown in Table 2, the texturization degree of the extrudates firstly increased and then decreased when the content of cysteine was between 0% and 0.25%. Specifically, the texturization degree of the extrudates increased from 2 to 2.35, illustrating that cysteine promoted the formation of fiber structure when the content of cysteine was between 0% and 0.15%. Compared with cysteine content of 0%, the texturization degree of the extrudates with cysteine content of 0.25% decreased to 1.51. It indicated that cysteine was not conducive to the formation of fiber structure when the content of cysteine exceeded 0.15%. The main reason might be that cysteine changed the cross-linking action of disulfide bonds between protein molecules. If excessive addition could affect the aggregation degree of protein molecules, it would have a negative impact on the formation of fiber structure during extrusion. Previous studies showed that the formation of fiber structure in extrudates was closely related to the protein–protein interaction [20,27].

The texture properties of the extrudates, including hardness, springiness and chewiness, are shown in Table 2. The hardness, springiness and chewiness of the extrudates firstly increased and then decreased when the content of cysteine was between 0% and 0.25%. When the addition of cysteine did not exceed 0.15%, cysteine could improve the hardness, springiness and chewiness of the extrudates. This might be because the excessive addition of cysteine affected the aggregation degree of protein molecules through the action of disulfide bonds, thus, affecting the texture properties of the extrudates. Therefore, it could be inferred that cysteine content higher than 0.15% had a negative impact on the extrusion, while the content of cysteine less than 0.15% could improve the formation of fiber structure and texture properties of the extrudates.

### 3.3. Microstructure

The micrographs of the extrudates obtained by scanning electron microscope are depicted in Figure 1. The internal fiber structure of the control sample (Figure 1a) could be clearly seen, because the protein molecules were decomposed, unfolded, aggregated, cross-linked and rearranged under the action of high temperature, high pressure and high shear during extrusion, resulting in fiber structure being formed [36]. The microstructure of the extrudates with 0.05%, 0.10% and 0.15% of cysteine content revealed more compact and clear fiber structure (Figure 1(b1–b3)). In contrast, the fiber structure of the extrudates with 0.20% and 0.25% of cysteine content became overall loose and contained more air (Figure 1(b4,b5)).

The result showed that the fiber structure in the extrudates was enhanced when the addition of cysteine was less than 0.15%, while the fiber structure became loose when the addition of cysteine exceeded 0.15%, which was consistent with the results of the texturization degree. A previous study indicated that the changes in chemical composition and protein molecular interactions might result in changes in microstructure of the extrudates [21]. This indicated that cysteine interfered with the cross-linking and rearrangement of protein molecules through disulfide bonds and affected the formation of fiber structure of the extrudates.

### 3.4. Rheological Properties

Frequency sweeps were measured in order to further explore the effect of cysteine on the viscoelasticity of the extrudates. The research was carried out on hydrated powdered extrudates. It is known that the rheological properties of dispersion could reflect the association and aggregation of protein molecules [7]. The results of the storage modulus (G′) and loss modulus (G″) are shown in Figure 2 G′ and G″ represented the properties of elasticity and viscosity, respectively. The values of G′ and G″ rose gradually with increasing frequency, inferring that the protein molecular chains were gradually unfolded during a short period of oscillation. In the frequency range of the test, the value of G′ was higher than G″, which suggested that the dispersed samples had a viscoelastic gel behavior and elasticity was dominant [37].

The protein molecular chains were depolymerized and unfolded under the action of high temperature, high pressure and high shear during the extrusion and a gel network structure would be formed in the cooling die area. However, hydrophobic interactions, hydrogen bonds and disulfide bonds were vital factors in forming the gel network structure [38]. The values of G′ and G″ firstly increased and then decreased with an increase in cysteine content, which might be affected by the mutual aggregation and cross-linking between protein molecules. In the extrusion process, when the content of cysteine was less than 0.15%, the protein molecular chains were unfolded and rearranged and the interaction between protein molecules was enhanced, which could easily form a strong gel. When the content of cysteine exceeded 0.15%, it would have a negative impact on the aggregation and cross-linking degree of protein molecules and an increase in G′ and G″ would be hindered, which easily formed a weak gel [39]. The result could be inferred that cysteine (<0.15%) could promote the formation of protein elastic gel through the cross-linking of disulfide bonds and excessive addition of cysteine (>0.15%) was not conducive to the formation of elastic gel.

### 3.5. Protein Solubility

The interaction between protein molecules was determined by protein solubility test. Table 3 exhibits the solubility of the plant-based extrudates with the addition of cysteine in four extraction solutions. According to the solubility in the four extraction solutions, the specific chemical forces between protein molecules were calculated [20,30]. (1): PB, protein molecules in the native state; (2)-(1): Urea, hydrogen bonds; (3)-(1): SDS, hydrophobic interactions; (4)-(1): 2-ME, disulfide bonds.

The amount of protein dissolved in four extraction solutions of the extrudate sample is shown in Table 3. The solubility of proteins was higher in Urea and SDS than in 2-ME extraction solutions, indicating that non-covalent bonds (hydrogen bonds and hydrophobic interactions) contributed greatly to the maintenance of protein conformation, followed by covalent bonds (disulfide bonds), which was consistent with the previous research conclusions [40,41]. With the increase in cysteine content, the content of protein solubility in urea, SDS and 2-ME extraction solutions firstly increased and then decreased, indicating that the hydrogen bonds, hydrophobic interactions and disulfide bonds, maintaining the intermolecular forces of proteins, firstly strengthened and then weakened. This was because cysteine had a reducing sulfhydryl group, which could form new disulfide bonds to change the intramolecular or intermolecular conformation of proteins, indirectly affecting the effect of non-covalent bonds on protein molecules and, thus, changing the degree of cross-linking, rearrangement and aggregation between protein molecules. Cysteine could destroy the disulfide bond that maintained the protein structure, expose the active group of the proteins and promote the formation of more new disulfide bonds and non-covalent bonds in the cooling die, increasing the cross-linking degree of protein molecules [42]. Moreover, cysteine was an unstable reductant that would break the protein structure and weaken the force between protein molecules when the addition amount exceeded 0.15%.

### 3.6. Secondary Structure

Fourier transform infrared spectroscopy is a common method to determine the secondary structure of proteins. The corresponding amide I band (1600–1700 cm^−1^) in the infrared spectrum was processed with Gaussian curve fitting and second derivative by using Peakfit software (version 4.12, SPSS Inc, Chicago, IL, USA) and the secondary structure of the extrudates was obtained, as shown in Figure 3. Different regions of the amide I band (1600–1700 cm^−1^) represent specific protein secondary structures: β-sheet (1618–1640 cm^−1^), random coil (1640–1649 cm^−1^), α-helix (1650–1660 cm^−1^) and β-turn (1660–1700 cm^−1^) according to a previous study [43].

The percentage contents of α-helix, β-sheet, β-turn and random coil are shown in Table 4. Compared to the extrudates without the addition of cysteine, the content of α-helix had no significant change with increasing cysteine content, but the content of β-sheet decreased, β-turn and the random coil structure increased. When the addition of cysteine was less than 0.15%, the β-sheet, β-turn and the random coil structure content had no significant change, which indicated that the secondary structure of protein in the extrudates after adding cysteine was stable. Previous studies suggested that α-helix and β-sheet were a relatively ordered and stable secondary structure, while β-turn and random coil were a relatively disordered secondary structure [44,45]. When the addition of cysteine exceeded 0.15%, the β-sheet content decreased and the β-turn and the random coil structure content increased, which showed that the secondary structure became disordered. The secondary structure of proteins could be converted to each other during extrusion, which was caused by the influence of non-covalent and covalent cross-linking between protein molecules [46,47]. Cysteine affected the cross-linking degree of protein structure by changing the disulfide bond of protein molecules. Cysteine would break the structure of protein molecules and had a negative impact when the addition exceeded 0.15%.

### 3.7. Odor Analysis

Cysteine had a unique odor and could have an important impact on flavor when used in food systems. Therefore, it was worth exploring the flavor changes of products by adding cysteine to high-moisture extrudates. Electronic nose is sensitive to volatile odors and widely used in various fields, such as food [48]. The method of discriminant factor analysis (DFA) was used to statistically analyze the data collected by the electronic nose and the differences in the odor of the extrudates with different addition amounts of cysteine were characterized, as shown in Figure 4a. The contribution rates of discriminant factor 1 (DF1) and discriminant factor 2 (DF2) were 78.82% and 10.92%, respectively, and the cumulative contribution rate was 89.74%, indicating that these two groups of factors basically contained all the odor information of the extrudates. The data points of the same sample had a high degree of aggregation on the horizontal and vertical coordinates, which showed that the tested sample had a high repeatability and stability. When the addition of cysteine was 0.20% and 0.25%, the response range of the extrudates crossed, indicating that they had similar odor characteristics. The six groups of samples were distributed in different quadrants and the fitting range between the samples had fewer intersections and the distribution had a certain regularity (Figure 4a). It was shown that the addition of cysteine had a significant difference in the odor properties of the extrudates.

According to the different response values of volatile substances in the sample on different sensors, the radar chart analysis was established, as shown in Figure 4b. The analysis of the radar chart further confirmed the DFA conclusions. Among the 17 sensors of the electronic nose, 6 L series sensors did not respond significantly, while 6 P and 5 T series sensors had significant responses. As shown in Figure 4b, the identification of volatile components of the extrudates with different cysteine content by electronic nose mainly depended on P40/2 (sensitive to hydrogen sulfide, chlorine and fluoride), P30/1 (sensitive to carbon oxygen compound, combustion product, ammonia and alcohol), PA/2 (sensitive to ethanol, ammonia and amine compounds) and P40/1 (sensitive to chlorid and fluoride) sensors [49,50]. This might be mainly because cysteine was a sulfur-containing compound, which was easily oxidized and decomposed to form aldehyde, ammonia or hydrogen sulfide under extrusion conditions of high temperature, high pressure and high shear [51].

## 4. Conclusions

The physicochemical properties of high-moisture extrudates prepared from plant protein were influenced by the addition of cysteine. The results demonstrated that the addition of cysteine (<0.15%) enhanced the formation of fiber structure and improved the texturization degree, microstructure, rheological properties, hardness, springiness and chewiness of the extrudates by changing the cross-linking and rearrangement of protein molecules through disulfide bonds. Cysteine (<0.15%) had reductive sulfhydryl groups and could change the disulfide bonds of protein structure, promote the formation of more new disulfide bonds and non-covalent bonds and increase the cross-linking degree of protein molecules. However, the excessive addition of cysteine (>0.15%) would break the cross-linking action of protein molecules and had a negative impact on the formation of fiber structure. The secondary structure changes in protein showed that the ordered β-sheet structures were decreased and the disordered β-turn and random coil structures were increased, because the structure of protein molecules was broken by the addition of cysteine (>0.15%) and became disordered. Analysis of volatile components by electronic nose showed the odor difference mainly existed in halides, hydrocarbons, sulfides and amines, which suggested that cysteine could enhance the odor properties of the extrudates. If we want to determine the changes and effects of specific flavor compounds, we need further research with the help of molecular sensory science. It is expected that this paper can provide some theoretical basis for improving the quality of high-moisture plant-based extrudates. In future research, we should pay more attention to the texture and flavor properties in plant-based meat substitutes.

## Figures and Tables

**Figure 1 foods-11-03109-f001:**
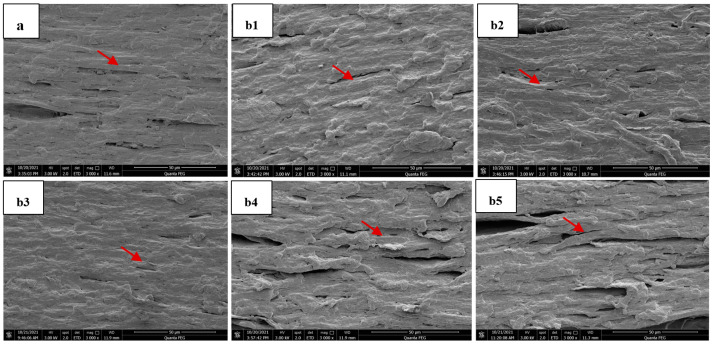
Effect of cysteine content on the microscopic morphological images of the extrudates. ((**a**): 0%; (**b1**–**b5**): 0.05%, 0.10%, 0.15%, 0.20% and 0.25%).

**Figure 2 foods-11-03109-f002:**
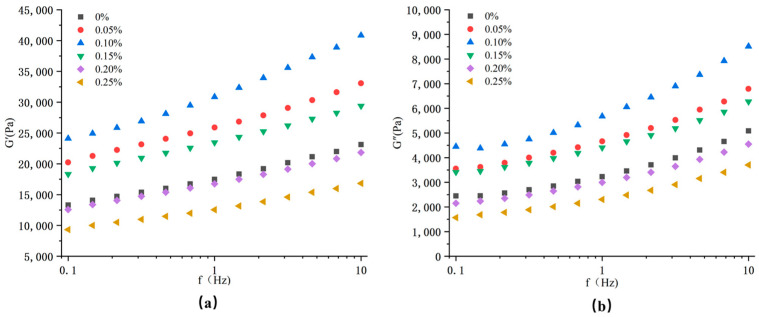
Storage modulus value G′(**a**) and loss modulus value G″ (**b**) value of the extrudates with the different addition of cysteine.

**Figure 3 foods-11-03109-f003:**
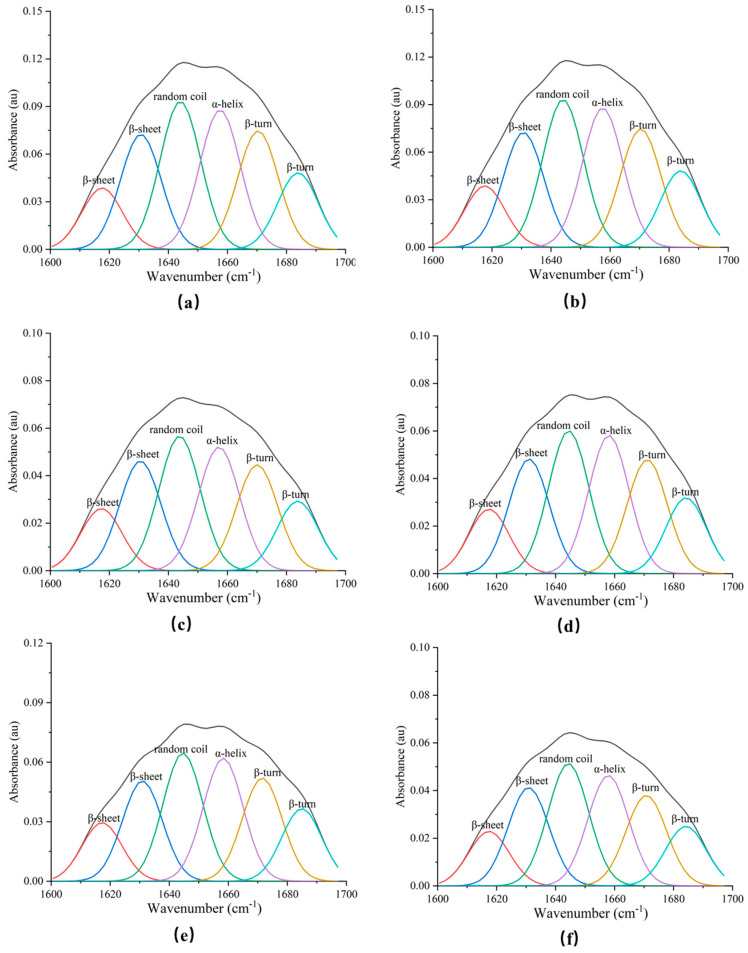
Fourier self-deconvoluted curve-fitted spectra for the extrudates with the different addition of cysteine ((**a**–**f**): 0%, 0.05%, 0.10%, 0.15%, 0.20% and 0.25%).

**Figure 4 foods-11-03109-f004:**
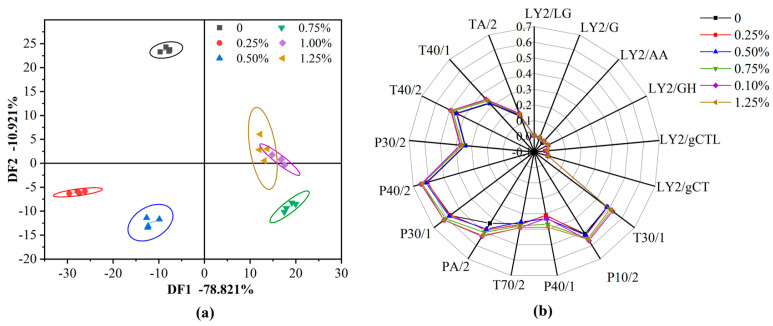
Discriminant factor (DF) analysis chart (**a**) and radar chart (**b**) of the extrudates with the different addition of cysteine.

**Table 1 foods-11-03109-t001:** Effects of cysteine content on the color of the extrudates.

Cysteine Content (%)	*L**	*a**	*b**	Δ*E**
0	60.77 ± 0.27 a	7.11 ± 0.17 a	11.45 ± 0.22 c	39.92 ± 0.19 b
0.05	59.81 ± 0.50 b	7.18 ± 0.10 a	11.69 ± 0.30 bc	40.90 ± 0.55 a
0.10	60.04 ± 0.35 ab	7.15 ± 0.15 a	11.69 ± 0.63 bc	40.69 ± 0.52 ab
0.15	59.78 ± 0.15 b	7.05 ± 0.08 ab	11.67 ± 0.62 bc	40.92 ± 0.34 a
0.20	60.05 ± 0.81 ab	6.90 ± 0.09 b	12.36 ± 0.42 ab	41.10 ± 0.74 a
0.25	59.61 ± 0.43 b	6.89 ± 0.11 b	12.66 ± 0.72 a	41.46 ± 0.35 a

Different lowercase letters in the same column represent significant differences (*p* < 0.05).

**Table 2 foods-11-03109-t002:** Effects of cysteine content on the texturization degree and texture properties of the extrudates.

Cysteine Content (%)	Texturization Degree	Hardness (kg)	Springiness (%)	Chewiness (kg)
0	2.00 ± 0.09 b	20.96 ± 0.31 ab	95.84 ± 2.29 b	17.59 ± 0.93 ab
0.05	2.35 ± 0.10 a	21.26 ± 0.16 a	97.11 ± 1.53 a	17.90 ± 0.36 a
0.10	2.23 ± 0.11 a	21.25 ± 0.64 a	97.66 ± 0.35 a	17.98 ± 0.57 a
0.15	2.24 ± 0.01 a	21.04 ± 0.32 ab	98.00 ± 0.03 a	17.52 ± 0.29 ab
0.20	1.80 ± 0.12 c	20.58 ± 0.20 b	91.50 ± 0.62 c	16.73 ± 0.20 b
0.25	1.51 ± 0.12 c	20.45 ± 0.38 b	91.79 ± 1.41 c	16.84 ± 0.29 b

Different lowercase letters in the same column represent significant differences (*p* < 0.05).

**Table 3 foods-11-03109-t003:** Changes in protein solubility of the extrudates with the different addition of cysteine induced by different extracting solution.

Cysteine Content (%)	Hydrogen Bonds (mg/g)	Hydrophobic Interactions (mg/g)	Disulfide Bonds (mg/g)
0	21.66 ± 0.20 e	18.55 ± 0.12 e	4.67 ± 0.01 bc
0.05	23.27 ± 0.25 d	23.47 ± 0.06 a	4.82 ± 0.07 b
0.10	26.84 ± 0.19 c	22.89 ± 0.05 b	5.75 ± 0.02 a
0.15	28.60 ± 0.09 a	22.44 ± 0.09 c	4.81 ± 0.07 b
0.20	27.67 ± 0.15 b	22.92 ± 0.03 b	4.60 ± 0.02 c
0.25	26.31 ± 0.07 c	21.82 ± 0.21 d	4.21 ± 0.02 d

Different lowercase letters in the same column represent significant differences (*p* < 0.05).

**Table 4 foods-11-03109-t004:** Effects of cysteine content on the relative percentage of protein secondary structure of the extrudates.

Cysteine Content (%)	α-Helix (%)	β-Sheet (%)	β-Turn (%)	Random Coil (%)
0	19.50 ± 0.07 ab	33.16 ± 0.14 a	28.21 ± 0.05 b	19.12 ± 0.01 c
0.05	19.58 ± 0.16 ab	32.67 ± 0.36 a	28.06 ± 0.06 b	19.70 ± 0.12 c
0.10	19.25 ± 0.07 b	32.45 ± 0.63 a	28.81 ± 0.64 ab	19.50 ± 0.18 c
0.15	19.83 ± 0.13 a	30.55 ± 0.29 b	29.73 ± 0.28 a	19.89 ± 0.44 b
0.20	19.50 ± 0.03 ab	30.54 ± 0.41 b	29.43 ± 0.31 a	20.53 ± 0.08 a
0.25	19.38 ± 0.18 b	30.01 ± 0.60 b	29.78 ± 0.17 a	20.83 ± 0.25 a

Different lowercase letters in the same column represent significant differences (*p* < 0.05).

## Data Availability

Data is contained within the article.

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
