# Peer review of "Effects of Cysteine on Physicochemical Properties of High-Moisture Extrudates Prepared from Plant Protein"

_foods, 2022, doi:10.3390/foods11193109_

Round 1

Reviewer 1 Report

The manuscript entitled "Effects of Cysteine on Physicochemical Properties of High Moisture Extrudates Prepared from Plant Protein" presented for review is a summary of the results of research on a very current issue related to the influence of various additives on the properties of textured proteins intended as meat substitutes.

The authors correctly interpreted obtained results and in most cases discussed them with the results of other studies (although the level of discussion, and sometimes the lack of it, is somewhat unsatisfied).

I have included my comments below:

1. Point 2.3. What is the purpose of calculating the ΔE between the white board and the individual samples? Especially since the authors did not provide any discussion of these results later in the paper? Isn't it better to calculate ΔE between 0% and the other variants? This will enable us to actually determine the color differentiation by adding cysteine.

2. In the methodology, two sections have been distinguished: Texturization degree determination and Texture properties determination in my opinion both describe texture tests with a different interpretation of the results. Anyway, in the discussion of the results, the authors combined both issues.

3. Please explain the purpose of the rheological research. If the aim of the study was to improve the fibrous structure, how will the results of the tests on the powdered preparation reflect into the assessment of the fibrous nature of the product?

4. Remove lines 180-182.

5. On what basis the authors found: “Moreover, the color of the extrudates significantly deepened and darkened when the addition of cysteine exceeded 0.15%”?

6. There is no description of ΔE results.

7. When describing the rheological properties, I disagree with the use of the term "extrudates". The research was carried out on hydrated powdered extrudates and such a concept should be included in the description.

8. Table 3. Please make the table design consistent with the others. Move the words Protein solubility over the other columns. Name the column I as Cysteine content.

9. Conclusion: line 382-385 should be included in the discussion rather than in the conclusions.

Author Response

Dear reviewer,

Thank you very much for your comments and professional advice. These opinions help to improve academic rigor of our article. The Manuscript (ID: foods-1882059) entitled "Effects of Cysteine on Physicochemical Properties of High Moisture Extrudates Prepared from Plant Protein" has been revised based on the reviewer’s comments and marked in red in the revised version. Thank you again for your kind consideration.

Reviewer 2 Report

The authors need to comment on these questions.

Line 61 to 64 – non-covalent bonds such as hydrophobic interactions and hydrogen bonds contribute to the formation and stabilizing of the structure of meat analogues, while SS bonds contribute to the formation of the fibrous structure. Please amend.

Line 125 – why do the authors lyophilize the extrudates even though they fixated with glutaraldehyde?  Lyophilisation of the sample might alter the structure as moisture was removed.

Figure 1 – the fibrous structures in the morphological images of extrudates are not obvious.

Line 136 – can the authors explain what is the rationale for using extrudate powder for rheological properties determination? How does the process of the extrudate to powder be useful to the study?

Line 247 – how does the hydration of extrudate powder to form a “gel” can be compared with the extrudate to analyze the rheological properties?

Line 336 and 337 – please give in 2 decimal places.

Author Response

(The authors gave the same response as above.)

Reviewer 3 Report

Review of “Effects of cysteine on physicochemical …. “ by Dai and An

The authors have investigated the effect of cysteine on various physicochemical properties of extruded  high moisture content mixture of soyprotein and wheat gluten. Various properties such as color, texture, microstructure, viscoelasticity, solubility and secondary structure. The results seem to indicate that all properties improve with an increase in cysteine concentration upto 0.15% and deteriorate at higher concentrations. The experimental methods are well described and the results well presented. On the whole, the manuscript is well written. The main criticism I have is that the authors have not explained clearly the mechanism by which cysteine controls the texture of the extruded product. The explanation that is provided in the manuscript is vague and therefore does not explain the existence of an optimum at a concentration of 0.15%. This needs to be addressed in the revision. Following are some specific comments.

1.      Line 207 - not clear. Need to explain why cross linking is decreased at cysteine concentration greater than 0.15%.

2.      Line 221 – why is there improvement in texture only upto 0.15%? is there a connection between texturization degree and texture? explain.

3.      Line 242 - need to comment on why the interference of cross linking by disulfide bond formation occurs only for concentrations above 0.15%

4.      Fig. 1a - one could see air pockets in the absence of cysteine (Fig. 1a). Comment.

5.      Line 260 -  authors state that protein molecules form a weak gel at cysteine concentrations above 0.15% though cysteine enhances gel strength at lower concentrations. Why?

6.      Section 3.5 needs more elaboration. It is not clear how different contributions (hydrogen bonds, hydrophobic interactions and disulfide bonds) in Table 3 were evaluated. Also, what are the units?

7.      Lines 290 – 295 – authors state “Cysteine could destroy the disulfide bond … when the addition amount exceeded 0.15%”. This section is speculative and not clear.

8.      Authors state that disulfide bonds do not play an important role (Table 3 and lines 282-284). Yet, they attribute the textural changes in the product at different cysteine concentrations to disulfide bonds. This needs to be elaborated.

Author Response

(The authors gave the same response as above.)

Round 2

Reviewer 2 Report

I have no other comments.

Reviewer 3 Report

Authors have incorporated the reviewers comments in their revised manuscript. Manuscript is acceptable for publication.